# Adequate immune response ensured by binary IL-2 and graded CD25 expression in a murine transfer model

**Franziska Fuhrmann[1,2†], Timo Lischke[2†], Fridolin Gross[3†], Tobias Scheel[2], Laura Bauer[1,2], Khalid Wasim Kalim[2], Andreas Radbruch[2,4], Hanspeter Herzel[3], Andreas Hutloff[1,2], Ria Baumgrass[2*]**

[1]Robert Koch Institute, Berlin, Germany; [2]German Rheumatism Research Center Berlin (DRFZ), A Leibniz Institute, Berlin, Germany; [3]Institute for Theoretical Biology, Charité University Medicine, Berlin, Germany; [4]Charité University Medicine, Berlin, Germany

**\*For correspondence:**
baumgrass@drfz.de

[†]These authors contributed equally to this work

**Competing interests:** The authors declare that no competing interests exist.

**Abstract** The IL-2/IL-2Ralpha (CD25) axis is of central importance for the interplay of effector and regulatory T cells. Nevertheless, the question how different antigen loads are translated into appropriate IL-2 production to ensure adequate responses against pathogens remains largely unexplored. Here we find that at single cell level, IL-2 is binary (digital) and CD25 is graded expressed whereas at population level both parameters show graded expression correlating with the antigen amount. Combining in vivo data with a mathematical model we demonstrate that only this binary IL-2 expression ensures a wide linear antigen response range for Teff and Treg cells under real spatiotempore conditions. Furthermore, at low antigen concentrations binary IL-2 expression safeguards by its spatial distribution selective STAT5 activation only of closely adjacent Treg cells regardless of their antigen specificity. These data show that the mode of IL-2 secretion is critical to tailor the adaptive immune response to the antigen amount.

## Introduction

The adaptive immune response of healthy vertebrates can be adjusted to match the overall severity of a pathogen. At the population level, T cells translate a total pathogen input into an adequate cytokine response over a large dynamic range. Individual T cells sense both antigen affinity and frequency and tailor the collective antigen-specific T cell activation to the antigen amount (*Allison et al., 2016*; *Gottschalk et al., 2012*; *Kellogg et al., 2015*; *Miskov-Zivanov et al., 2013*). The cytokine IL-2 and its signaling are of central importance for these processes. IL-2 is mainly produced by activated Foxp3[−] CD4[+] T helper cells (hereinafter referred to as 'Th cells') and mediates diverse pleiotropic actions such as promotion of Th1 and Th2 differentiation, expansion of multiple Th cell populations, as well as generation and maintenance of Foxp3[+] regulatory T cells (Treg cells) (*Liao et al., 2013*). Thus, IL-2 positively acts on Th cells through synergistic effects between antigen and cytokines and negatively through preferred binding to and activation of Foxp3[+] Treg cells.

There are a number of in vitro studies, some of them even combined with mathematical modeling, to dissect the dichotomic effect of IL-2 signaling in effector Th cells and Foxp3[+] Treg cells (*Benary et al., 2008*; *Bendfeldt et al., 2012*; *Busse et al., 2010*; *Feinerman et al., 2010*; *Tkach et al., 2014*; *Voisinne et al., 2015*). Observations from various knock-out mice (e.g. IL-2Rα, STAT5) and other in vivo studies, however, have challenged some conclusions of these in vitro data and underlined the importance of local antigen concentrations, temporal aspects, cytokine

compositions, cell heterogeneity in tissues and much more for the biological outcome orchestrated by IL-2 during an immune response (reviewed in *Liao et al., 2013*).

There are several articles dealing with certain molecular features of the IL-2 pathway in vivo, such as: First, Sojka et al. studied the kinetics of IL-2 secretion by naive and memory Th cells (*Sojka et al., 2004*) and some studies observed a paracrine IL-2 action on STAT5 activation (*Long and Adler, 2006*; *O'Gorman et al., 2009*; *Sabatos et al., 2008*). Second, Amado et al. proposed that IL-2 is sensed by both the activated Th cell pool and by Treg cells, which reciprocally regulate cells of the IL-2-producing cell subset in IL-2 reporter mice (*Amado et al., 2013*). Third, O'Gorman et al. showed that in the initial phase of antigen-specific activation IL-2-dependent STAT5 phosphorylation occurred primarily in Treg cells (*O'Gorman et al., 2009*). Fourth, Liu et al. underscored the impor-tance of a fine-tuned local IL-2 availability to limit otherwise constant damage from activated auto-responsive Th cells under steady state conditions (*Liu et al., 2015*).

However, in all of these studies the antigen load-dependent IL-2 expression and IL-2 signaling via CD25/STAT5 in both antigen-specific as well as endogenous Th and Treg cells was not continuously monitored in vivo. Furthermore, there is the open question whether a graded antigen load is trans-lated into binary (also known as digital, all-or-none, switch-like, or bimodal) or graded expression of TCR-dependent genes coding for important cytokines and receptors such as IL-2 and CD25 or CD69, respectively, in vivo. Even in vitro there were conflicting data about binary or graded expres-sion of CD25 and CD69 (*Allison et al., 2016*; *Busse et al., 2010*; *Feinerman et al., 2010*; *Huang et al., 2013*; *Podtschaske et al., 2007*; *Tkach et al., 2014*). This is an interesting issue because binary responses are able to filter out noisy signals and can integrate different information to control cell responses. Thereby, binary responses allow parallel and independent control of single cell activation probability and population heterogeneity (*Kellogg et al., 2015*; *Köck et al., 2014*).

Here, we studied the antigen amount-driven regulation of IL-2 and CD25 expression at the single cell and population level as well as IL-2 signaling under the real spatial-temporal conditions of sec-ondary lymphoid organs in vivo. Using a murine adoptive transfer model enabled us to study an anti-gen-specific T cell response in syngeneic recipient mice and cytokine secretion assays to measure the rate and magnitude of cytokine production of individual cells over time. In addition, mathematical modeling allowed us to compare the effects of the hypothesized graded and observed binary IL-2 expression for the first time. The model revealed advantages of the latter for a wide linear scaling of STAT5 activation in Treg and/or Th cells to antigen concentration. Finally, our results demonstrate that the combination of binary IL-2 secretion and graded CD25 expression at the cellular level ena-bles the parallel control of single cell activation probability and population heterogeneity by the respective antigen amount.

## Results

### Binary IL-2 production of Th cells in vivo after immunization

Previously, we showed in vitro that IL-2 is produced in a binary ('digital' or 'all-or-none') manner by Th cells (*Podtschaske et al., 2007*), because variation of strength of the stimulation changed the number of IL-2 expressing cells rather than the intensity of IL-2 per cell. It is challenging to analyze whether this holds true in vivo, too.

To address this question, we adoptively transferred naïve transgenic Foxp3$^-$ CD4$^+$ T helper cells from DO11.10 mice into recipient mice and analyzed IL-2 production in the spleen using a cytokine secretion assay (*Assenmacher et al., 1998*) 14 hr after immunization with ovalbumin (OVA). Different amounts of antigen used for immunization resulted in changes in the number of IL-2-producing cells (*Figure 1A*, left), but almost no change in the mean fluorescence intensity (MFI) of IL-2 expression per Th cell (*Figure 1A*, right). Thus, the amount of antigen correlates with the proportion of Th cells that produced IL-2, but does not change the amount of IL-2 per cell, which confirms that IL-2 pro-duction is regulated in a binary fashion in vivo, too. The IL-2 expression peaked between 6 and 18 hr (*Figure 1D*). The IL-2 secretion assay detected up to 75% IL-2-producing Th cells at the highest amount of antigen (2 mg) used. This number reflects the in vivo situation better than direct measure-ment of IL-2 by intracellular staining, which only detected maximally ~9% IL-2-producing Th cells, considered to be too low (*Long and Adler, 2006*). Moreover, we excluded an underestimation of cytokine production by confirming that the cytokine capture matrix is not saturated in our

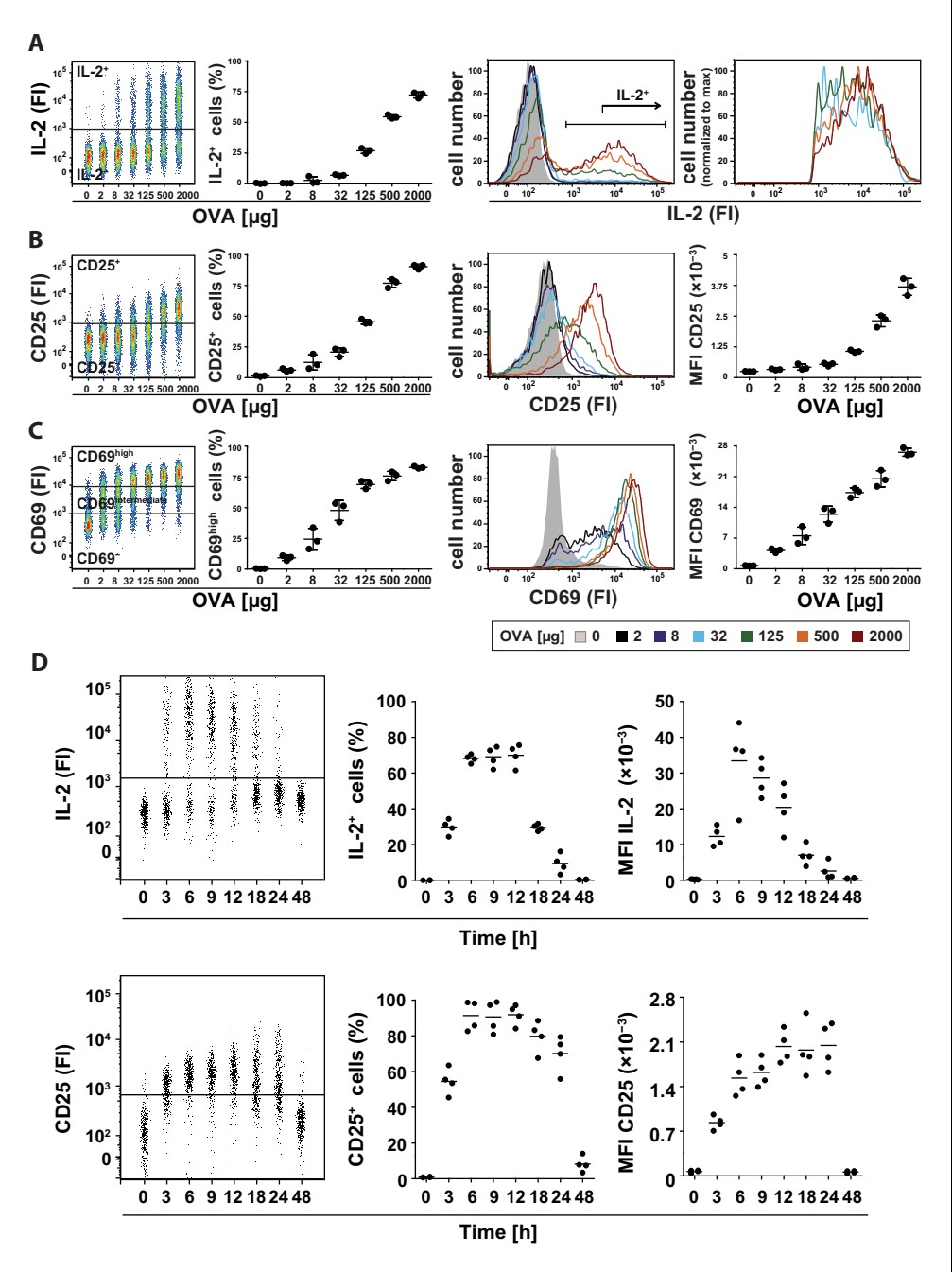

**Figure 1.** Binary IL-2 secretion of adoptively transferred Th cells in vivo after immunization with increasing amounts of antigen and kinetics of IL-2 secretion. BALB/c mice, adoptively transferred with OVA-specific T cells from DO11.10 mice, were immunized intravenously with the indicated amounts of OVA and 10 μg LPS as adjuvant and analyzed 14 hr later (A–C) or with 2 mg OVA + 10 μg LPS and analyzed over time (D). IL-2 secretion of OVA-TCR[+] CD4[+] T cells was analyzed using the IL-2 secretion assay. Data from gated transgenic T cells (live B220[−] CD4[+] OVA-TCR[+] Foxp3[−]) were concatenated (3 mice per antigen dose). (A) IL-2 production was plotted against the amount of OVA used for immunization: Very left dot plot: fluorescence intensities of IL-2 per cell; left graph: frequencies of IL-2 producing cells; right histogram overlay: IL-2 fluorescence intensities (FI) for all gated OVA-TCR[+] CD4[+] T cells; very right histogram overlay: IL-2 producing OVA-TCR[+] CD4[+] T cells. (B) Similar analysis for the expression of CD25 on gated OVA-TCR[+] CD4[+] T cells. In the very right graph the mean fluorescence intensity (MFI) of CD25 is plotted against the amount of OVA. (C) Similar analysis for the expression of CD69 on gated OVA-TCR[+] CD4[+] T cells. CD69 expressing T cells were distinguished in antigen-specific activated CD69[high] and bystander-activated CD69[intermediate] expressing T cells. The gates for CD69[intermediate] and CD69[high] T cells were set

*Figure 1 continued on next page*

*Figure 1 continued*

according to unimmunized and non-transfer immunized controls. (D) IL-2 production (top row) and CD25 expression (bottom row) in OVA-TCR$^+$ CD4$^+$ T cells are plotted against time after immunization. Dot plots (left) depict fluorescence intensities per cell; the middle graphs depict frequencies of positive cells; the right graphs depict MFIs. Statistics: mean and standard deviation were plotted in all graphs. Data are representative of three independent experiments.

The following figure supplements are available for figure 1:

**Figure supplement 1.** The cytokine capture matrix is not saturated by the endogenous IL-2 secretion.

**Figure supplement 2.** Binary IL-2 secretion of adoptively transferred OT-II Th cells in vivo after immunization with increasing amounts of antigen.

**Figure supplement 3.** Confirmation of binary IL-2 expression using Gaussian mixture models.

**Figure supplement 4.** Transcriptional regulation of IL-2 secretion in adoptively transferred Th cells.

**Figure supplement 5.** Expression levels of CD25 and CD69 on T cell subpopulations after immunization.

**Figure supplement 6.** Expression levels of CD25 on T cell subpopulations at rest and after immunization.

experiments (*Figure 1—figure supplement 1*). The data shown were generated with transfer of OVA-specific DO11.10 T cells (*Figure 1*), but were also reproduced using T cells from OT-II mice (*Figure 1—figure supplement 2*). Our results were confirmed in a more thorough statistical analysis in which we explicitly tested for bimodality in IL-2 expression and used mixed gaussian models to extract expression values and percentages of cell subpopulations (*Figure 1—figure supplement 3*).

In parallel we studied whether the amount of the transcription factors c-Fos and NFATc2 limit the IL-2 production per mouse cell in vivo, as already shown in vitro for human peripheral blood memory Th cells (*Podtschaske et al., 2007*). To this end we used adoptively transferred naïve transgenic CD4$^+$ T cells 7 hr after immunization of mice and co-stained c-Fos, NFATc2, and IL-2 using intracellular staining and cytokine secretion assays, respectively. Interestingly, only the amount of c-Fos but not of NFATc2 per Th cell correlates with the probability of IL-2 expression in vivo. Quartiles with the same cell number but different fluorescence intensities of c-Fos or NFATc2 (*Figure 1—figure supplement 4*, left) showed that the probability of IL-2 production correlates with the c-Fos amount per cell but is independent of NFATc2 amounts within their physiological range (*Figure 1—figure supplement 4*, right). This is in line with the observation of Allison et al. that the MEK/ERK/AP1 axis plays an important role in translating TCR signaling into proportional activation of genes essential for T cell function (*Allison et al., 2016*).

## Graded IL-2Rα expression by Th cells in vivo after immunization

Next we analyzed how CD25, the high affinity IL-2Rα chain, is expressed in vivo. So far, in vitro quantitative single cell studies have produced conflicting results proposing opposite manners of CD25 expression, namely binary (*Busse et al., 2010*) and graded (*Feinerman et al., 2010*). Given that CD25 expression peaks between 12 and 24 hr after immunization (*Lischke et al., 2012*; *Long and Adler, 2006*), we observed graded CD25 and CD69 expression of adoptively transferred transgenic CD4$^+$ T cells 14 hr after immunization of mice. Different antigen concentrations used for immunization changed the mean fluorescence intensity (MFI) of CD25 expression per Th cell (*Figure 1B*, right). The amount of antigen correlated with the amount of CD25 expressed per cell (*Figure 1B*, left), and thus its ability to respond to IL-2. Notably, CD25 expression was up-regulated only by transferred antigen-specific Th and Treg cells (*Figure 1—figure supplements 5A* and *6*), but not by endogenous T cells. This fact is pointing to preferential up-regulation of CD25 expression via TCR-stimulated T cell activation. The same holds true for high expression of CD69 (*Figure 1—figure supplement 5B*). However, a small bystander expression of CD69 (CD69$^{intermediate}$ cells) was observed on endogenous Th and Treg cells after immunization of mice with OVA + LPS.

Next, we analyzed the kinetics of IL-2 secretion and CD25 expression in adoptively transferred transgenic Th cells after immunization of mice with 2 mg OVA + 10 µg LPS (*Figure 1D*). Both processes reached their maximum value between 3 to 6 hr and obviously CD25 protein expression persisted longer than IL-2 secretion.

## CD25 upregulation is similar in IL-2-producing and IL-2-nonproducing transgenic T cells in vivo

The amount of expression of CD25 is critical for Th cells in order to respond to IL-2, in particular if the IL-2 concentration is low (*Busse et al., 2010*; *Feinerman et al., 2010*; *Popmihajlov and Smith, 2008*; *Tkach et al., 2014*). In contrast to Treg cells, Th cells show high expression of CD25 only transiently, due to the short duration of both, antigen-induced TCR signaling and IL-2/IL-2R signaling (*Malek and Castro, 2010*; *Tkach et al., 2014*).

Interestingly, co-staining of CD25 and IL-2 in adoptively transferred transgenic Th cells (*Figure 2A*, top row) revealed that the amount and variation of CD25 expression is similar in IL-2-producing and IL-2-nonproducing cells 14 hr after immunization with different antigen concentrations (*Figure 2A*, bottom row). Accordingly, the kinetics of IL-2 secretion and CD25 expression in adoptively transferred transgenic Th cells after immunization of mice with 2 mg OVA + 10 µg LPS was similar in IL-2-producing and IL-2-nonproducing cells during the time of expression of both molecules (*Figure 2B*). These data suggest that mainly the amount of antigen and a paracrine, rather than an autocrine, action of IL-2 regulates the expression of CD25 in vivo.

## Computational modeling suggests physiological advantages of binary IL-2 secretion

To get an idea about the physiological importance of binary IL-2 expression for activation of antigen-specific Th and Treg cells, we compared the observed binary IL-2 pattern to the hypothetical scenario of graded IL-2 secretion. The physiological consequences of different secretion patterns are not intuitively obvious and might depend on many factors such as the spatial distribution of cells in the spleen and on diffusion and secretion rates. Computational modeling allowed us to simulate the consequences of different hypothetical scenarios taking into account quantitative empirical information about the real system. All parameters of the model are listed in *Supplementary file 1* and the spatial distribution of T cells is depicted in *Figure 3—figure supplement 1*.

Our computational model describes a population of T cells that can secrete IL-2 and/or express IL-2R at varying levels corresponding to the antigen stimulus. Cells are considered 'activated' when the amount of receptor-bound IL-2 exceeds a particular threshold. *Figure 3A* schematically represents the differences between the compared scenarios at low antigen concentrations. Intuitively, one might expect that activation of Th cells requires higher levels of antigen stimulus in the graded scenario because for IL-2 diffusion it is harder to overcome the critical distances between cells. It is not obvious, however, whether there are differences in the activation of Treg cells as they are able to sense very low concentrations of IL-2. The model can assist in making sense of or in correcting our intuitions.

Our simulations show that in the binary scenario for IL-2 the number of activated Th and Treg cells slowly increases with increasing antigen stimulus, while the behaviour is rather switch-like in the graded case (*Figure 3B*). This can be understood in the following way: If secretion is graded (i.e. many producers), the distribution of IL-2 is relatively homogeneous across the whole volume. Consequently, all cells are exposed to similar levels of IL-2 and thus are either all activated or not in the graded IL-2 scenario. If secretion is binary (i.e. the number of secreting cells increases with increasing stimulus, the amount is constant), by contrast, then the number of activated cells increases more gradually because cells that are in close proximity to secreting cells are activated first. Binary secretion at the cellular level thus leads to graded activation at the population level and *vice versa*.

In summary, binary IL-2 and graded CD25 expression as well as higher CD25 expression by Treg cells ensures a fast action and wide linear antigen response range of Treg cells. Interestingly, the model simulations predict that Th cells are activated at lower antigen concentrations in the binary than in the graded scenario. However, Treg cells are always activated at lower antigen concentrations than Th cells. This is obviously important for effective control of autoimmunity ensuring Treg

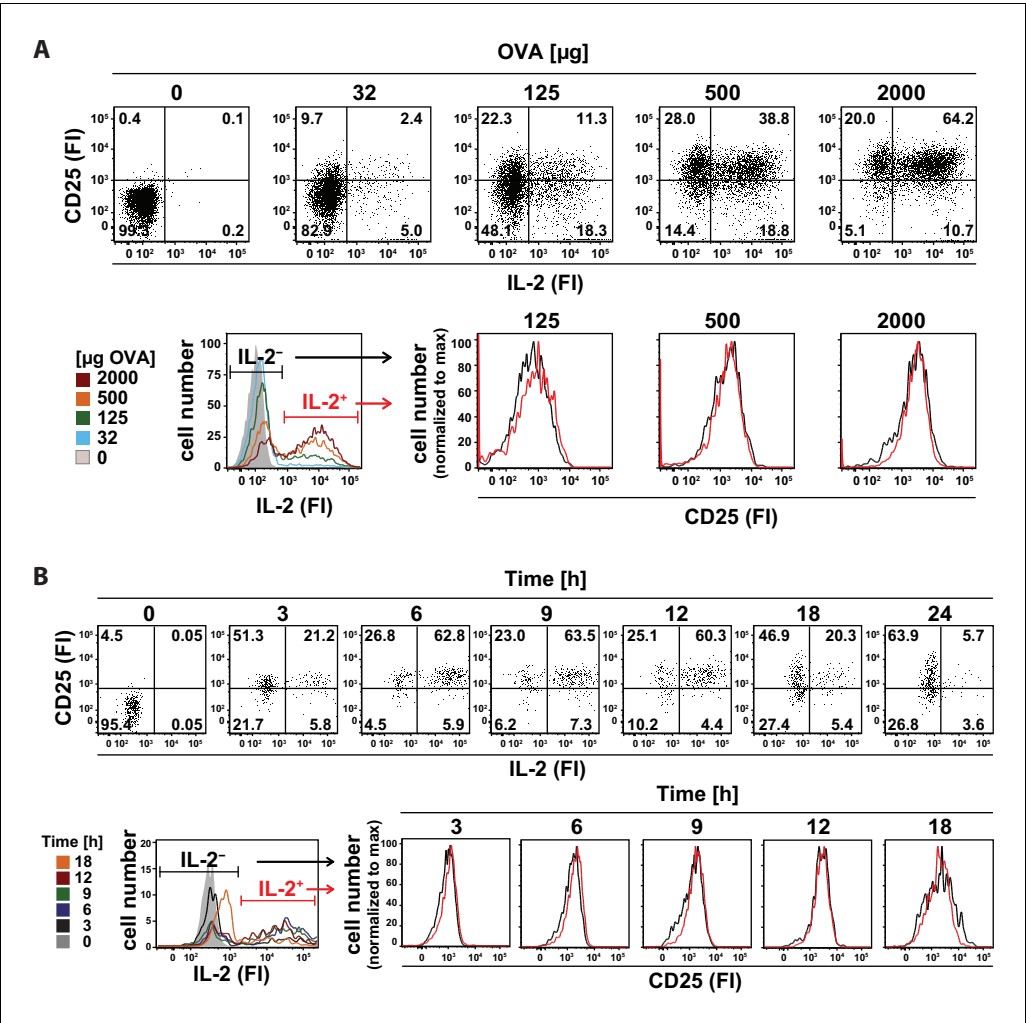

**Figure 2.** In vivo kinetics of IL-2 secretion and CD25 expression in adoptively transferred Th cells after immunization and their correlation upon titration of antigen amount as well as over time. BALB/c mice, adoptively transferred with OVA-specific T cells from DO11.10 mice, were immunized intravenously with increasing amounts of OVA and 10 µg LPS as adjuvant and analyzed at 14 hr (**A**) or 2 mg OVA + 10 µg LPS and analyzed over time (**B**). IL-2 secretion of OVA-TCR+ CD4+ T cells was analyzed using the IL-2 secretion assay. Data from gated OVA-TCR+ CD4+ T cells (live B220− CD4+ OVA-TCR+ Foxp3−) were concatenated (3 mice per antigen dose or time point). (**A**) IL-2 production and CD25 expression (fluorescence intensities per cell) are depicted for increasing amounts of OVA used for immunization (top row, dot plots). A histogram overlay shows the IL-2 fluorescence intensities of gated OVA-TCR+ CD4+ T cells for different antigen amounts for direct comparison (bottom row, left). Histogram overlays of CD25 fluorescence intensities in IL-2 producing (red) and non-producing (black) OVA-TCR+ CD4+ T cells for individual OVA amounts are shown for direct comparison of CD25 expression (bottom row, right). (**B**) IL-2 production and CD25 expression (fluorescence intensities per cell) are depicted over time (top row, dot plots). A histogram overlay shows the IL-2 fluorescence intensities of gated OVA-TCR+ CD4+ T cells over time for direct comparison (bottom row, left). Histogram overlays of CD25 fluorescence intensities in IL-2 producing (red) and non-producing (black) OVA-TCR+ CD4+ T cells for individual time points are shown for direct comparison of CD25 expression (bottom row, right). Statistics: mean and standard deviation were plotted in all graphs. Data are representative of three independent experiments.

cell activation in spatial proximity even if only rare self-reactive IL-2 secreting Th cells exist (*Liu et al., 2015*).

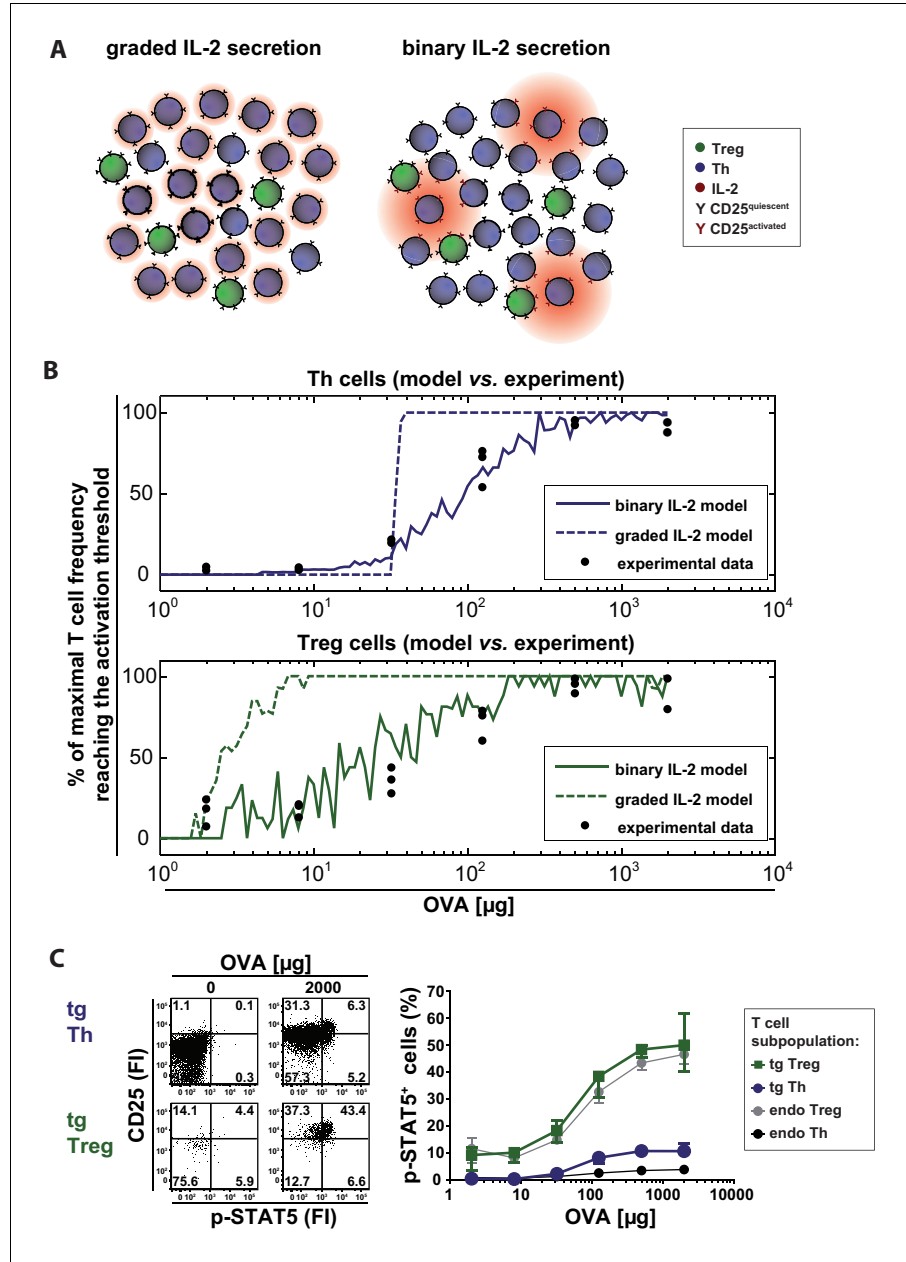

**Figure 3.** Model and experimental validation of binary versus graded IL-2 production. (**A**) Schematic representation for the effects of different modes of IL-2 secretion (red) by Th (blue) on surrounding Th (blue) and Treg (green) cells. Graded IL-2 secretion at low antigen dose (left) is compared to binary IL-2 secretion (right). IL-2 mediated T cell activation is depicted via conversion of quiescent CD25 (black Y) to activated CD25 (red Y). (**B**) Model and comparison to experiment: Diffusion of IL-2 and interaction of IL-2 and IL-2R is modeled for a population of T cells in the spleen. The figure depicts the fraction of T cells that have more than a critical amount IL-2 bound to IL-2R on their surface (and thus reach their activation threshold) and compares it to the fraction of p-STAT5$^+$ T cells in the experiment. (**C**) BALB/c mice, adoptively transferred with OVA-specific T cells from DO11.10 mice, were immunized intravenously with increasing amounts of OVA and 10 µg LPS as adjuvant. CD25 expression and STAT5 phosphorylation were analyzed after 14 hr on gated CD4$^+$ T cells (live B220$^-$ CD4$^+$ OVA-TCR$^{+/-}$ Foxp3$^{+/-}$). Left figure: Dot plots of p-STAT5 versus CD25 expression for gated transgenic Th and Treg cells without immunization or after immunization with 2 mg OVA + 10 µg LPS. Right graph: Percentage of p-STAT5$^+$ signal in different T cell subpopulations versus amount of OVA used for immunization (3 mice per antigen dose). Statistics: mean and standard deviation were plotted. Data are representative of two independent experiments.

*Figure 3 continued on next page*

*Figure 3 continued*

The following figure supplement is available for figure 3:

**Figure supplement 1.** Simulation model of binary versus graded IL-2 production.

## Binary IL-2 secretion tailors the number of activated Th and Treg cells to the amount of antigen

To compare the model data with the in vivo situation, we analysed IL-2 signaling by STAT5 activation in antigen-specific T cells. We co-stained phosphorylated STAT5 and CD25 directly ex vivo in adoptively transferred antigen-specific Th and Treg cells 14 hr after immunization of mice. As expected, with increasing amounts of antigen (and hence increasing IL-2 secretion) STAT5 phosphorylation increased proportionally in antigen-specific Th and Treg cells. Moreover, antigen-specific Treg cells showed a much higher STAT5 phosphorylation than antigen-specific Th cells. However, the p-STAT5 signals were quite low. Even at the highest antigen concentration (2 mg OVA) only ~11% of Th cells stained positive for p-STAT5 despite the fact that about 75% of Th cells secrete IL-2 and about 95% express CD25 at this condition (*Figure 3C*). Within the Treg cell population about 50% were p-STAT5[+] (*Figure 3C*). It is possible that p-STAT5[+] frequencies are underestimated due to difficulties in preserving the p-STAT5 signal during the preparation of cells ex vivo and the staining process. O' Gorman et al. described that the induction of p-STAT5 is lower in naïve Th cells compared to repeatedly activated cells but lasts longer (*O'Gorman et al., 2009*). In line with this, our endogenous antigen-unspecific Th cells had a very low expression of CD25 and almost no p-STAT5. In contrast, STAT5 activation (*Figure 3C*) was observed at a similar degree on both Treg cell populations, transgenic (antigen-specific) and endogenous Treg cells.

To analyse whether the low p-STAT5 levels of antigen-specific Th cells are due to their unresponsiveness to IL-2 or because they do not have access to IL-2 in vivo, we treated mice with either recombinant IL-2 (20 µg) or IL-2-complex (2.5 µg IL-2 with 10 µg anti-IL-2 antibody JES6-5H4). Furthermore, we incubated one part of the spleen in vitro with IL-2 (200 ng recombinant IL-2 for 25 min). The data clearly show that neither IL-2 treatment of mice nor in vitro IL-2 treatment of Th cells increased STAT5 activation in transgenic Th cells (*Figure 4*). Thus it is very probable that antigen-specific Th cells are unresponsive to IL-2 by negative crosstalk between antigen and IL-2 signaling as described before (*Tkach et al., 2014*; *Villarino et al., 2007*; *Waysbort et al., 2013*). Taken together, our model data are consistent with the experimental data (*Figure 3C*) and reveal the importance of IL-2-activated Treg cells for limiting the immune response of Th cells. First, at low amounts of antigen (<30 µg OVA) preferentially a proportion of Treg cells is activated by IL-2, and thus can limit the immune response at low danger. Second, binary IL-2 expression tailors the fraction of activated Treg cells proportionately to the antigen amount and avoids that all Treg cells are almost simultaneously activated as would be expected in the graded IL-2 expression scenario. Third, at low antigen amounts binary IL-2 expression ensures by its spatial distribution that only those antigen-specific and unspecific Treg cells are activated, which are in close spatial proximity to the few IL-2 producing Th cells. This is in good agreement with data of Liu et al. concerning observations on self tolerance (*Liu et al., 2015*).

## The ratio of antigen-specific Th and Treg cells remains stable over a wide antigen concentration range

Next, we investigated whether the observed preferential Treg cell activation at low antigen doses is translated into a higher expansion or survival rate compared to Th cells. Using the adoptive transfer model, we found that antigen-specific Th and Treg cells did not proliferate after immunization at low antigen concentrations (2–8 µg OVA) in CFSE dilution assays (*Figure 5A*). In line with this, we observed similar total cell counts and relative frequencies as in the unimmunized controls under these conditions (*Figure 5B,C*). The percentage of dividing progenitors was approximately 10%, which appeared to be the background level already present in Th and Treg cells without immunization (*Figure 5D*). Above 8 µg OVA, total cell numbers of transgenic Th and Treg cells increased proportionally to the antigen amount, but with a dramatic different slope. CFSE dilution profiles revealed a stronger proliferative response of OVA-specific Th cells compared to Treg cells at higher

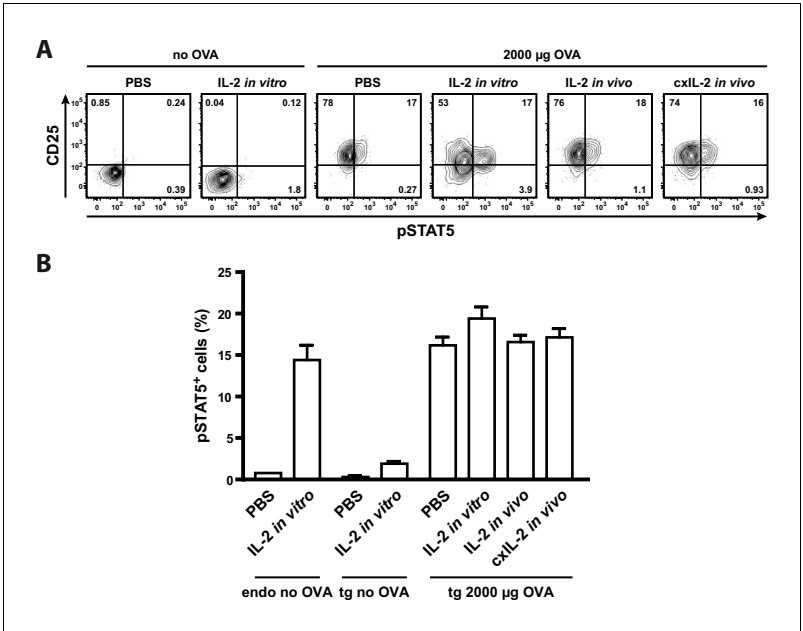

**Figure 4.** Treatment of mice with recombinant IL-2 does not increase STAT5 activation. C57BL/6 mice, adoptively transferred with OVA-specific T cells from OT-II mice, were immunized intravenously with 2000 μg OVA and 10 μg LPS or were left unimmunized. After 12 hr mice received either 20 μg recombinant IL-2, 2.5 μg IL-2 complexed with 10 μg anti-IL-2 (JES6-5H4; targets IL-2 to CD122), or PBS as control. Mice were sacrificed 1 hr later. Cells from one part of the spleen were incubated for 25 min in vitro with 200 ng recombinant IL-2. (**A**) Representative original data (gated on transgenic CD4[+] T cells) are shown in the flow cytometric plots. (**B**) Percentage of p-STAT5[+] cells within either endogenous CD4[+] T cells (including FoxP3[+] cells) or transgenic FoxP3-negative CD4[+] T cells are depicted in bar graphs. Pooled data are from two independent experiments (together six mice per group). Error bars represent SEM.

concentrations. As an example, at 500 μg OVA 94% of Th cell progenitors were dividing compared to 65% of Treg cell progenitors (*Figure 5D*).

Subsequently, we analyzed the apoptosis rate of transferred antigen-specific Th and Treg cells 72 hr after immunization using annexin V binding (a marker for loss of cell membrane integrity upon apoptosis) (*Vermes et al., 1995*) and FLICA staining (detecting activated caspases) (*Thornberry et al., 1997*). In both assays, antigen-specific Treg cells showed less apoptosis than antigen-specific Th cells, in particular at low antigen concentrations (4–5% and 7–9% apoptotic cells, respectively, at 2–8 μg OVA). Apoptosis of Treg cells was proportionally increasing with the antigen amount up to the level of Th cells at 2000 μg OVA (*Figure 5—figure supplement 1*). In contrast, the apoptosis rate of antigen-specific Th cells was almost constant over the covered antigen concentration range.

Comparing cell counts at low antigen concentrations (2–8 μg OVA) (left parts of *Figure 5B* and *Figure 5—figure supplement 1*), it is obvious that apoptosis does not play an important role because T cell numbers were only slightly reduced at 72 hr. However, one must pay attention to the fact that apoptosis data represent only a momentary picture in the course of the immune response whereas proliferation analyses (CFSE dilution assay) are reflecting the whole process. In fact, the dilution of CFSE upon cell divisions reveals the proliferative history of the adoptively transferred T cells and reflects the total amount of proliferation that has occurred in the course of the immune response.

Altogether, the initial response to an antigen is characterized by antigen-specific IL-2 production of Th cells and an early activation of antigen-specific and endogenous Treg cells (*Figures 1* and *3*). At low antigen concentrations (up to 8 μg OVA) there is: (i) activation of Th and Treg cells, but (ii) almost no proliferation of all cells because neither Treg nor Th cells reach the activation threshold for entry into it, and (iii) a slightly lower apoptosis rate of Treg cells compared to Th cells. At higher

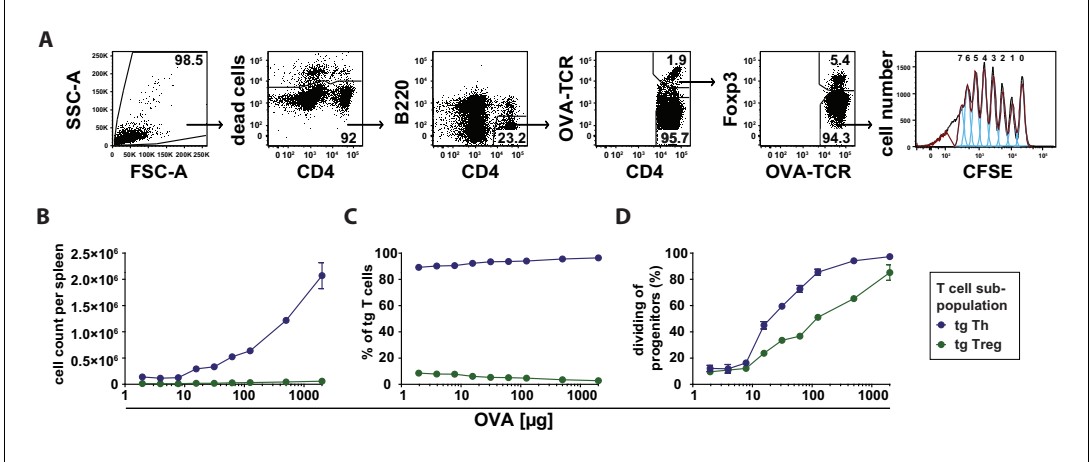

**Figure 5.** Dependency of in vivo proliferation rate of adoptively transferred Th and Treg cells on amount of antigen. BALB/c mice, adoptively transferred with CFSE-labeled OVA-specific T cells from DO11.10 mice, were immunized intravenously with increasing amounts of OVA and 10 µg LPS as adjuvant and analyzed at 72 hr. Gated OVA-TCR$^+$ CD4$^+$ T cells (live B220$^-$ CD4$^+$ OVA-TCR$^+$ Foxp3$^{+/-}$) were analyzed for CFSE dilution (2 mice per antigen dose). (**A**) Gating strategy for proliferation analysis: after doublet exclusion in FSC-A/FSC-H and SSC-A/SSC-H plots (not shown), cells were gated according to scatter characteristics, dead cells (Pacific Orange positive) were excluded, followed by gating on B220$^-$ CD4$^+$ OVA-TCR$^+$ Foxp3$^{+/-}$ T cells and analysis of CFSE dilution. (**B**) The total T cell count per spleen was calculated and plotted against amount of antigen for OVA-TCR$^+$ Foxp3$^-$ Th cells (blue circles) and OVA-TCR$^+$ Foxp3$^+$ Treg cells (green circles). (**C**) The relative frequencies of Foxp3$^-$ Th cells (blue circles) and Foxp3$^+$ Treg cells (green circles) in all transgenic OVA-TCR$^+$ T cells were plotted against amount of antigen. (**D**) Furthermore, the percentage of dividing cells of the progenitor cells (generation 0) was calculated and plotted against the amount of OVA used for immunization. Statistics: mean and standard deviation were plotted in all graphs. Data are representative of four independent experiments.

The following figure supplement is available for figure 5:

**Figure supplement 1.** Antigen-specific Treg cells show less apoptosis then conventional Th cells at low amounts of antigen.

concentrations (>8 µg OVA) the cells expand proportionately to the antigen load, but it is obvious that living antigen-specific Th expand more than Treg cells (*Figure 5B–D*), which is in contrast to a previous report using IL-2 reporter mice (*Amado et al., 2013*).

## Discussion

In order to contribute to the main question, how the adaptive immune system ensures tolerance at very low antigen concentration and adequate immune response at rising antigen amounts in respect of IL-2 expression and IL-2 action, we asked here: Is a graded antigen load translated into a binary or graded expression of the cytokine IL-2, the IL-2 receptor CD25, and the activation marker CD69 shortly after immunization of mice? Do binary processes indeed contribute to an enhanced robustness (*Kellogg et al., 2015*) of antigen-specific activation of Th cells? Do binary processes support antigen amount-dependent Treg cell-mediated tolerance? Is IL-2 acting in a paracrine as well as autocrine fashion on CD25 expression?

Binding of IL-2 influences CD25 expression in Th cells as shown by several groups using single cell in vitro experiments including mathematical modeling (*Busse et al., 2010*; *Feinerman et al., 2008*; *Tkach et al., 2014*). IL-2 enhances TCR-induced up-regulation of CD25 level at suboptimal stimulation (*Feinerman et al., 2008*). Our data clearly demonstrate a paracrine and ruled out a sole autocrine in vivo action of IL-2. Co-staining of CD25 and IL-2 revealed that IL-2-producing and IL-2-non-producing transgenic CD4$^+$ T cells have the same expression of CD25 at all the different antigen concentrations used (*Figure 2A*) and over the whole time frame of IL-2 and CD25 co-expression (*Figure 2C*). These data are consistent with previous reports showing via STAT5 activation that IL-2 is delivered to neighboring T cells via synapses (*Sabatos et al., 2008*) and that IL-2 production does not coincide with STAT5 phosphorylation (*Long and Adler, 2006*). The observed peak times of IL-2

(6–14 hr) and CD25 (12–24 hr) show that IL-2 expression precedes CD25 expression in vivo, which is in good agreement with previous observations (*Lischke et al., 2012*; *Long and Adler, 2006*).

Beside the delivery of an important survival and maintenance signal for Treg cells, only a paracrine action of IL-2 allows in addition at least two other most recently discovered processes to occur. First, weaker T cell clones can be activated and proliferate just by integrating antigen and IL-2 response. Thus, IL-2 from strongly activated clones co-opt a fraction of weaker clones into activation and proliferation (*Voisinne et al., 2015*) and contribute thereby to continuous heterogeneity of the responding population. Second, IL-2 may act as a gatekeeper for T cell homeostasis during an immune response, initiating the down-modulation of excessive or boosting of weak effector T cell responses by the interplay of Th and Treg cells (*Almeida et al., 2012*, *2006*; *Amado et al., 2013*; *Reynolds et al., 2014*).

In vitro studies claimed that many readouts downstream of TCR-activation such as IL-2, CD69, CD25, and ppERK are always binary (*Feinerman et al., 2010*; *Tkach et al., 2014*). However, there are conflicting data from in vitro experiments concerning CD69 and CD25 expression (*Allison et al., 2016*; *Busse et al., 2010*; *Feinerman et al., 2010*; *Kimmig and Baumgrass, 2009*; *Podtschaske et al., 2007*; *Tkach et al., 2014*). Using the adoptive transfer approach, we could confirm binary IL-2 expression in vivo, because the quantities of this cytokine on the cellular level were independent of the antigen concentrations used (*Figure 1A*). However, we also found that the overall IL-2 responses were not binary but graded at the population level. Increasing antigen amount resulted in a proportional higher percentage of IL-2-producing T cells. Thus, increasing or decreasing the number of IL-2 producers and not regulating the magnitude of IL-2 output of individual cells calibrates the antigen response. Interestingly, unlike IL-2, and in contrast to some in vitro data, CD25 and CD69 expression is clearly graded in vivo at both, single cell and population level (*Figure 1B,C*) (*Feinerman et al., 2010*; *Tkach et al., 2014*).

Binary responses of T cells play not just a role in filtering out noisy signals but moreover they can integrate information, such as timing or intensity, of the danger signal to adapt cell responses (*Kellogg et al., 2015*). On the other hand, as discussed recently by Tkach et al., "binary single parameter readouts carry some limitations: they are prone to saturation of input detection range, exhibit narrow output dynamic ranges, are subject to shifts in value by cellular death and migration, and provide little information on the functional capacity of activated cells" (*Tkach et al., 2014*).

Therefore, we asked whether the interplay of binary IL-2 and graded CD25 expression might have physiological advantages. To this end, we generated a computational model to compare the potential consequences of the observed binary and a hypothetical graded IL-2 secretion for the adaptive immune response. The model allowed us to simulate the outcome of these different scenarios taking into account quantitative spatial information in the spleen (*Busse et al., 2010*; *Chapman et al., 1981*; *Lischke et al., 2012*; *Mueller and Germain, 2009*; *Sabatos et al., 2008*; *Smith, 2006*). Our model confirmed that Treg cells become activated by IL-2 at lower antigen concentrations than Th cells and predicted that graded IL-2 secretion would result in activation of all T cells in a very narrow range of antigen stimulus, which is represented by a very steep slope of the activation curve of Th and Treg cells. Moreover, the binary IL-2 secretion results in a broadened linear range of initial T cell activation. This suits the needs of a fine-tuned immune response for a broad range of possible antigen loads.

Crucially, p-STAT5 measurements could substantiate our modeling results. Binary IL-2 secretion of Th cells led to a graded antigen dose-dependent increase in STAT5 phosphorylation in antigen-specific Th and Treg cells in vivo, which correlated perfectly with the activation curve of antigen-specific Th and Treg cells in our computational model. Very remarkably, not only the number of antigen-specific Treg cells but to a similar degree also endogenous Treg cells are p-STAT5 positive and correlate with the antigen amount. However, it is known that not only IL-2 signaling but also TCR signaling is required for the Treg cells to gain and sustain their full suppressive activity (*Levine et al., 2014*; *Vahl et al., 2014*).

Recently, Liu et al. provided information on how the spatial organization of IL-2-producing Th cells and activated Treg cells might help to maintain immune homeostasis (*Liu et al., 2015*). Using imaging flow cytometry (histo-cytometry) they discovered that STAT5 was phosphorylated in most clustering Treg cells and that an IL-2-producing Th cell was usually at the center of Treg cell clusters. This study convincingly highlighted the role of spatial proximity of few IL-2 producing Th cells and clustering of Treg cells close to them for critical IL-2-driven immune regulatory processes. The

reduction of motility of Th and Treg cells within the clusters could secure effective IL-2 delivery for Treg cells even if IL-2 is released in a non-directional manner. Moreover, the number of Treg cells within the cluster might determine how many Treg cells are affected. The point, how IL-2 is shared between Th and Treg cells is extensively studied and discussed in two recent theoretical articles (*Labowsky, 2016*; *Thurley et al., 2015*).

Almeida et al. deduced a central role for IL-2 in homeostasis of Th cells by indexing of Treg cells to the number of IL-2 producing cells resulting in a constant ratio of both in IL-2 reporter mice (*Almeida et al., 2006*; *Amado et al., 2013*). In contrast, we showed in our transfer model a higher increase in the number of antigen-specific Th cells compared to Treg cells over the antigen concentration range used (*Figure 5B–D*). Nevertheless, the interdependence of IL-2 producing cells and Treg cells allows immune responses but also feedback control.

Most recently, Chinen et al. proved the essential role of IL-2R-dependent activation of STAT5 in controlling suppressor function of mature Treg cells (*Chinen et al., 2016*). Using genetic gain and loss of function approaches they demonstrated that augmented STAT5 activation in differentiated Treg cells increases the formation of Treg-DC-cell-conjugates and potentiates suppressor function in a TCR-independent manner.

In summary, we have shown that there are substantial differences between in vitro and in vivo experimental data concerning IL-2 and CD25 expression and regulation. Furthermore, we have demonstrated that computational modeling serves as a valuable and indispensable tool to understand complex regulatory mechanisms in vivo. Our insights contribute to a better understanding of antigen dose-dependent activation of the IL-2/CD25 axis of Th and Treg cells in an in vivo context and therefore under the real spatio-temporal conditions.

## Materials and methods

### Mice

OVA-TCR transgenic OT-II mice (Jackson Laboratory, Bar Harbor, ME, Stock 004194) were crossed to B6PL mice (Jackson Stock 000406). OVA-TCR transgenic DO11.10 mice (Jackson Stock 003303) were for some experiments additionally crossed with DEREG mice (*Lahl et al., 2007*) to obtain GFP reporter mice for Foxp3. All these mice including non-transgenic C57BL/6NCrl (Charles River, Sulzfeld, Germany, Strain Code 027) were bred under specific-pathogen-free conditions in the animal facility of the Federal Institute for Risk Assessment (Berlin, Germany). BALB/cAnNCrl mice were purchased from Charles River (Strain Code 028).

### Adoptive transfer

T cell receptor transgenic cells were isolated from spleens according to standard procedures. Naive OT-II cells were enriched by positive MACS sort with L-Selectin (CD62L) beads (Miltenyi Biotec, Bergisch Gladbach, Germany) and transferred into C57BL/6 mice ($2.5$–$3.0 \times 10^6$ transgenic T cells per recipient). DO11.10 cells were depleted of $CD8^+$ cells by negative MACS sort with CD8 beads (Miltenyi Biotec) and transferred into BALB/c mice ($1.0$–$1.5 \times 10^6$ transgenic T cells per recipient). To analyze proliferation, cells were labeled with CFSE (Invitrogen) before adoptive transfer, according to standard protocols. The mice were immunized intravenously with different amounts of endotoxin-free Ovalbumin (<5 pg of endotoxin/mg of protein by Limulus amebocyte lysate assay) and 10 µg lipopolysaccharide (LPS, from *E. coli* O55:B5, Sigma-Aldrich, St. Louis, MO, USA) as adjuvant. For IL-2 in vivo treatment, mice received i.p. either 20 µg recombinant IL-2 or 2.5 µg IL-2 complexed with 10 µg anti-IL-2 (clone JES6-5H4 targets IL-2 to CD122). Splenocytes from C57BL/6 or BALB/c recipient mice were isolated and analyzed at indicated times after immunization.

### IL-2 secretion assay

IL-2 secretion assays (Miltenyi Biotech) were performed with $7.5 \times 10^6$ splenocytes from C57BL/6 mice as previously described (*Assenmacher et al., 1998*). In brief, cells were washed, resuspended in ice-cold buffer (0.5% BSA, 2 mM EDTA in PBS) and incubated with the anti-IL-2 antibody capture matrix on ice for 5–10 min. After adding pre-warmed medium (RPMI 1640 + 4% FCS), cells were incubated at 37°C in a shaking water bath for 45 min. The secretion phase was stopped by adding ice cold PBS/BSA/EDTA buffer to the cells. IL-2 secreting cells were stained with anti-IL-2-PE

antibody and different surface marker antibodies for 20 min on ice. To control whether the IL-2 capture matrix is saturated by endogenous IL-2 secretion, 0.4 µg/ml recombinant murine IL-2 (eBioscience) was added 5 min before the end of the secretion phase (see *Figure 1—figure supplement 1*).

## Flow cytometric analysis

Cell suspensions from spleen were counted with a Guava EasyCyte capillary flow cytometer and Via-Count Assay (Merck Millipore, Germany). For reduction of unspecific antibody binding, cells were preincubated with 100 µg/ml 2.4G2 (FcγRII/III; ATCC, Manassas, VA) and 50 µg/ml purified rat Ig (Nordic, Tilburg, The Netherlands) for 5 min. Surface staining was performed on ice for 20 min with monoclonal antibodies conjugated to FITC, PE, PerCP-Cy5.5, PE-Cy7, APC, AF700, and PacB: αCD4 (RM4-5), αCD25 (PC61.5), αB220 (RA3-6B2), αOVA-TCR (KJ1-26), αThy1.1 (OX-7; all purified from hybridoma supernatants), and αCD69 (H1.2F3; eBioscience, San Diego, CA, RRID:AB_465119). Dead cells were excluded from analysis by DAPI (4-,6-diamidino-2-phenylindole) staining.

For detection of early apoptotic cells, either PE-conjugated annexin V (BioLegend, RRID:AB_2561298) or an Alexa Fluor 660-labeled inhibitor of caspases (FLICA; Immunochemistry Technologies) was used in combination with DAPI.

When analyzing CFSE dilution profiles the percentage of dividing progenitors, i.e. the precursor frequency of dividing cells, was calculated as described (*Lyons and Parish, 1994*).

For intracellular staining of the transcription factor p-STAT5 (Alexa Fluor 647-labeled antibody clone 47/p-STAT5, BD Biosciences) spleens were disintegrated directly into BD Fixation Buffer containing 2% paraformaldehyde and stained in BD Perm Buffer III (BD Biosciences) for 30 min on ice.

For intracellular staining of the transcription factors NFATc2 (Cy5-labeled own polyclonal antibodies (*Bendfeldt et al., 2012*), c-Fos (Alexa Fluor 488-labeled rabbit polyclonal IgG antibodies from Santa Cruz Biotechnology, RRID:AB_2231996), and Foxp3 (PE-Cy7-labeled antibody clone FJK-16s from eBioscience, RRID:AB_891554) the cells were stained with 1.34 µM Pacific Orange succinimidyl ester to exclude dead cells, fixed with Foxp3 fixation buffer (eBioscience), stained in Foxp3 permeabilization buffer (eBioscience), and analyzed using a LSR II flow cytometer (BD Biosciences). Data were analyzed with FlowJo software (Treestar, Ashland, OR).

## Computational model
### Basic features of the model

The model describes a population of T cells on a 2D circular domain (spleen/lymph node) and explicitly includes the secretion and diffusion of IL-2 and the formation of IL-2:IL-2R-complexes in space and time via a set of reaction-diffusion equations. Cells are modeled as circular areas with reactive boundaries that account for either secretion of IL-2 or IL-2:IL-2R-complex formation. Dirichlet boundary conditions were imposed on the outer boundary of the domain. The cells were distributed randomly with a mean cell-to-cell distance derived from considerations about the number of T cells and the size of the T cell zone in the spleen. The dynamics of IL-2:IL-2R-interactions on the cell boundaries was included in terms of a simple Michaelis-Menten type model which accounts for complex formation, dissociation and internalization. We assumed that the receptor dynamics can be approximated by considering only the influence of the expression of the high affinity chain CD25. The influence of unspecific CD25 positive cells was modeled in terms of a homogeneous background degradation of IL-2. The model was written and evaluated in MATLAB making use of the Partial Differential Equation Toolbox.

### Implementation of binary/graded scenarios

Binary (all-or-none) IL-2 secretion was implemented by randomly choosing a subset of producers that all secrete IL-2 at the maximal rate. In the graded scenario, by contrast, the maximum amount of producers secrete IL-2 at a rate that changes with increasing antigen stimulus. The fraction of producers in the binary case and the secretion rate in the graded case were chosen such that the total amount of IL-2 secreted is the same for both scenarios. The quantitative relationship between IL-2 secretion rate/CD25 expression and the antigen stimulus were determined by evaluating the corresponding FACS experiments (*Figure 1*).

## Output of the model

Simulations were performed for varying antigen stimuli according to the doses used in the corresponding FACS experiments. For each stimulus the process was simulated over a time span of 48 hr which encompasses the initial increase and eventual decline of both IL-2 secretion and CD25 expression. In each simulation the distribution of cells and the random selection of producers/consumers was carried out anew. For each consumer cell the total amount of internalized complex was measured over the course of the simulation and taken as a proxy for downstream STAT5 signaling (not included in the model). We assumed that a cell is activated when the amount of complex exceeds a critical threshold. This threshold was chosen such as to optimize the fit between the fraction of activated cells in the model and the fraction of p-STAT5$^+$ cells in the experiment (**Figure 3C**).

## Equations

The basic scheme of the diffusion equation used in the model is

$$\frac{\partial u}{\partial t} = D\frac{\partial^2 u}{\partial x^2} - dUR \cdot IL2R \cdot \frac{u}{u+k},$$

where $u$ represents the solution for the concentration of IL-2 and is the diffusion coefficient. The second term on the right hand side accounts for the homogeneous background degradation due to unspecific regulatory cells. *IL2R* is the expression rate of CD25 and the factor *dUR* stands for the density of unspecific Tregs. The parameter $k$ can be interpreted as the concentration at which the consumption of IL-2 is half its maximum value. This parameter depends on the rates of formation, dissociation, and internalization of complexes in the following way:

$$k = \frac{k_{deg} + k_{off}}{k_{on}}.$$

The secretion of IL-2 by producer cells was accounted for in terms of generalized Neumann conditions on the outer cell boundaries:

$$n \cdot (D\nabla u) = \frac{IL2 \cdot f(A,t)}{cCell},$$

where $n$ is the outward unit normal, *IL2* the maximal secretion rate, and *cCell* the cell circumference. The factor $f(A,t)$ accounts for the dependence of the secretion rate on the antigen stimulus and the time after stimulation. Dependence on time was modeled as the composition of a linearly increasing function from 0 to the maximum value between t = 0 and t = 9 hr and a linearly decreasing function from the maximum to 0 between t = 9 hr and t = 18 hr (compare to **Figure 3**).

The interaction of IL-2 and IL-2R on the surface of consumer cells is also accounted for in terms of generalized Neumann boundary conditions:

$$n \cdot (D\nabla u) = \frac{-IL2R \cdot g(A,t)}{cCell} \cdot \frac{u}{u+k}.$$

Here, *IL2R* denotes the maximal expression of CD25, while $g(A,t)$ accounts for the dependence of CD25 expression on stimulus and time. Time dependence in this case was modeled as a linear increase between t = 0 and t = 18 hr and a subsequent decrease between t = 18 hr and t = 30. The dependence of CD25 expression on the antigen stimulus is different for T helper cells and regulatory T cells and was determined by evaluating the corresponding FACS experiments.

## Acknowledgements

We thank Tim Sparwasser, Jochen Hühn, Ute Hoffmann and Alf Hamann for providing DO11.10×DEREG mice. We are grateful to the staff of the animal facility and the labmanagers. We thank Jenny Kirsch and Toralf Kaiser, operators of the flow cytometry core facility (FCCF). We thank Melanie Venzke, Christian Gabriel, Manja Jargosch, Anna Abajyan, Martin Karl, Katharina Hecklau, Stefanie Gryzik and Yen Hoang for lively discussion and helpful advices.

## Additional information

### Funding

| Funder | Grant reference number | Author |
|---|---|---|
| Bundesministerium für Bildung und Forschung | T-Sys (e:Bio) | Andreas Radbruch<br>Hanspeter Herzel<br>Ria Baumgrass |
| Deutsche Forschungsgemeinschaft | SFB650 | Ria Baumgrass |
| Deutsche Forschungsgemeinschaft | TR52 | Ria Baumgrass |

The funders had no role in study design, data collection and interpretation, or the decision to submit the work for publication.

### Author contributions

FF, Conception and design, Acquisition of data, Analysis and interpretation of data; TL, FG, Conception and design, Acquisition of data, Analysis and interpretation of data, Drafting or revising the article; TS, LB, KWK, Acquisition of data, Analysis and interpretation of data; AR, Conception and design, Drafting or revising the article; HH, AH, RB, Conception and design, Analysis and interpretation of data, Drafting or revising the article

### Author ORCIDs

Timo Lischke, http://orcid.org/0000-0003-0413-4252
Hanspeter Herzel, http://orcid.org/0000-0003-0414-7889
Ria Baumgrass, http://orcid.org/0000-0002-3289-1608

### Ethics

Animal experimentation: All animal experiments were performed according to state guidelines and approved by the responsible governmental authority LAGeSo (Landesamt für Gesundheit und Soziales) Berlin in animal experiment licenses T 0187/01 and G 0070/13.

## Additional files

### Supplementary files

• Supplementary file 1: Parameter estimates for the mathematical model. Parameter names, symbols and values used for the mathematical modeling are given.

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
