## [Decision Letter]

Thank you for submitting your article "Adequate immune response ensured by binary IL-2 and graded CD25 expression in vivo" for consideration by *eLife*. Your article has been favorably evaluated by Tadatsugu Taniguchi (Senior Editor) and three reviewers, one of whom, Michael L Dustin (Reviewer #1), is a member of our Board of Reviewing Editors. The following individual involved in review of your submission has agreed to reveal their identity: Kendall A Smith (Reviewer #3).

The reviewers have discussed the reviews with one another and the Reviewing Editor has drafted this decision to help you prepare a revised submission.

Summary:

This paper sets out to clarify how T cells scale their response to MHC-peptide dose at the level of IL-2 production, IL-2 response, IL-2 consumption, Treg activation and proliferation in vivo. It provides a culmination of >35 years of research on how the cytokine IL-2 and its receptor make a profound contribution to shaping the T cell response. They re-evaluate the in vivo dose response and time course of IL-2 production, CD25 expression, CD69 expression and proliferation in vivo using DO11.10 and OTII TCR transgenic systems. They claim that IL-2 production is binary (see below)- that as antigen dose increases, the proportion of T cells making a fixed amount of IL-2 over several hours increases. In contrast, CD25 and CD69 responses are reported that all responding T cells make a response proportional to the input antigen dose. While experiments like this have been done before, the results for both IL-2 production and CD25 expression have been reported to be binary or graded in different studies so it was important to clarify the results in vivo using well defined models. They then apply a 2D model the T cell zone to the data to better understand the mechanism by which the combination of a binary IL-2 production and graded IL-2 responsiveness combine to generate a proliferative response in Th that is proportional to the antigen input. To make their case, they use a previously-published computational model of IL-2 dynamics [1] to resolve this paradox: by controlling the number of cells that secrete IL-2 in a graded manner, the system avoids homogenization, adjusts the number of Treg involved in suppressing the response, and establish a nice analog response. Such illuminating insight comes at the time when immunologists are dwelling on issues of scaling and global regulation: Fuhrmann et al. provide a very thorough analysis (both from the experimental and theoretical point of view).

Essential revisions:

1) The analysis of the digitalness of T cell activation in the experiment is partially satisfactory. The data speak for themselves (Figure 1) but a better analysis of the data would apply Gaussian Mixture Models to extract the mode of IL-2 secretion or CD25 expression amongst the activated cells. A more complete statistical analysis to demonstrate when/how a dual vs. single mode of activation applies (cf Figure 2 in [2]).

2) One issue that this article seems to stumble open is about the distribution of pSTAT5 in response to IL-2 (cf Figure 3). Fuhrmann et al. report that the frequency of STAT5 phosphorylating cells is very small in conditions when a large fraction of cells is secreting IL-2. This can be well accounted for when taking into consideration the reported negative crosstalk between antigen signaling and IL-2 signaling [3-6]. Such negative crosstalk has already been invoked to explain some of the paradoxical features of STAT5 activation in T cells [3, 4], as well as scaling in IL-2 secretion. One control experiment would help clarify this issue: injecting a large dose of IL-2 or IL-2 complexed with a stimulatory antibody, then monitoring STAT5 phosphorylation would distinguish whether T cells are unresponsive to IL-2 or do not have access to IL-2. Blocking TCR signaling (e.g. using a blocking antibody against CD4) would also resolve whether there is an antigen-dependent active block on IL-2 signaling in vivo.

3) Throughout the manuscript, antigen "strength" is referred to, but the reader is left with no molecular definition of this mysterious "strength". Since these experiments were performed with TCR transgenic cells, each cellular TCR is monoclonal. Thus, the affinity between the TCR and pMHC complexes is identical on all cells. Consequently, to account for the [antigen] log/dose response curve, isn't the differences on the part of the responses seen due to a log/normal distribution of [TCR] per cell?

References:

1) Busse, D., et al., Competing feedback loops shape IL-2 signaling between helper and regulatory T lymphocytes in cellular microenvironments. Proc Natl Acad Sci U S A, 2010. 107(7): p. 3058-63.

2) Shalek, A.K., et al., Single-cell RNA-seq reveals dynamic paracrine control of cellular variation. Nature, 2014. 510(7505): p. 363-9.

3) Long, M. and A.J. Adler, Cutting edge: Paracrine, but not autocrine, IL-2 signaling is sustained during early antiviral CD4 T cell response. J Immunol, 2006. 177(7): p. 4257-61.

4) Villarino, A.V., et al., Helper T cell IL-2 production is limited by negative feedback and STAT-dependent cytokine signals. J Exp Med, 2007. 204(1): p. 65-71.

5) Waysbort, N., et al., Coupled IL-2-dependent extracellular feedbacks govern two distinct consecutive phases of CD4 T cell activation. J Immunol, 2013. 191(12): p. 5822-30.

6) Tkach, K.E., et al., T cells translate individual, quantal activation into collective, analog cytokine responses via time-integrated feedbacks. *ELife*, 2014. 3: p. e01944.

---

## [Author Response]

*Essential revisions:*

*1) The analysis of the digitalness of T cell activation in the experiment is partially satisfactory. The data speak for themselves (Figure 1) but a better analysis of the data would apply Gaussian Mixture Models to extract the mode of IL-2 secretion or CD25 expression amongst the activated cells. A more complete statistical analysis to demonstrate when/how a dual vs. single mode of activation applies (cf Figure 2 in [2]).*

As recommended, we performed an analysis based on Gaussian mixture models to extract the modes of expression and percentages of activated IL-2 cells. We tried several options to demonstrate bimodality (model selection using AIC/BIC, Silverman test, dip test). Only the dip test statistic provided satisfactory results, whereas the other methods seemed too sensitive, for example detecting significant mulitmodality due to spurious modes in the data. We used mixture models with two components only if the test revealed significant bimodality.

We inserted the data as a novel supplementary figure (Figure 1—figure supplement 3: Confirmation of binary IL-2 expression using Gaussian mixture models) and a paragraph into the manuscript.

*2) One issue that this article seems to stumble open is about the distribution of pSTAT5 in response to IL-2 (cf Figure 3). Fuhrmann et al. report that the frequency of STAT5 phosphorylating cells is very small in conditions when a large fraction of cells is secreting IL-2. This can be well accounted for when taking into consideration the reported negative crosstalk between antigen signaling and IL-2 signaling [3-6]. Such negative crosstalk has already been invoked to explain some of the paradoxical features of STAT5 activation in T cells [3, 4], as well as scaling in IL-2 secretion. One control experiment would help clarify this issue: injecting a large dose of IL-2 or IL-2 complexed with a stimulatory antibody, then monitoring STAT5 phosphorylation would distinguish whether T cells are unresponsive to IL-2 or do not have access to IL-2. Blocking TCR signaling (e.g. using a blocking antibody against CD4) would also resolve whether there is an antigen-dependent active block on IL-2 signaling* in vivo.

We took the reviewer’s suggestion and provided additional experiments to shed some light into this issue. We included a novel figure (Figure 4) and a paragraph into the manuscript:

“To analyze whether the low pSTAT5 levels of antigen-specific Th cells are due to their unresponsiveness to IL-2 or because they do not have access to IL-2 in vivo, we treated mice with either recombinant IL-2 (20 µg) or IL-2-complex (2.5 µg IL-2 with 10 µg anti-IL-2 antibody JES6-5H4). […] Thus it is very probable that antigen-specific Th cells are unresponsive to IL-2 by negative crosstalk between antigen and IL-2 signaling as described before (Tkach et al., 2014; Villarino et al., 2007; Waysbort et al., 2013).”

*3) Throughout the manuscript, antigen "strength" is referred to, but the reader is left with no molecular definition of this mysterious "strength". Since these experiments were performed with TCR transgenic cells, each cellular TCR is monoclonal. Thus, the affinity between the TCR and pMHC complexes is identical on all cells. Consequently, to account for the [antigen] log/dose response curve, isn't the differences on the part of the responses seen due to a log/normal distribution of [TCR] per cell?*

We thank the reviewer for the advice and changed it into “amount of antigen”.

*References:*

*1) Busse, D., et al., Competing feedback loops shape IL-2 signaling between helper and regulatory T lymphocytes in cellular microenvironments. Proc Natl Acad Sci U S A, 2010. 107(7): p. 3058-63.*

*2) Shalek, A.K., et al., Single-cell RNA-seq reveals dynamic paracrine control of cellular variation. Nature, 2014. 510(7505): p. 363-9.*

*3) Long, M. and A.J. Adler, Cutting edge: Paracrine, but not autocrine, IL-2 signaling is sustained during early antiviral CD4 T cell response. J Immunol, 2006. 177(7): p. 4257-61.*

*4) Villarino, A.V., et al., Helper T cell IL-2 production is limited by negative feedback and STAT-dependent cytokine signals. J Exp Med, 2007. 204(1): p. 65-71.*

*5) Waysbort, N., et al., Coupled IL-2-dependent extracellular feedbacks govern two distinct consecutive phases of CD4 T cell activation. J Immunol, 2013. 191(12): p. 5822-30.*

*6) Tkach, K.E., et al., T cells translate individual, quantal activation into collective, analog cytokine responses via time-integrated feedbacks. ELife, 2014. 3: p. e01944.*